# Antimicrobial Susceptibility to 27 Drugs and the Molecular Mechanisms of Macrolide, Tetracycline, and Quinolone Resistance in *Gemella* sp.

**DOI:** 10.3390/antibiotics12101538

**Published:** 2023-10-14

**Authors:** Michiko Furugaito, Yuko Arai, Yutaka Uzawa, Toshinori Kamisako, Kohei Ogura, Shigefumi Okamoto, Ken Kikuchi

**Affiliations:** 1Department of Clinical Laboratory and Biomedical Sciences, Graduate School of Medicine, Osaka University, Suita, Osaka 565-0871, Japan; michiko-furugaito@med.kindai.ac.jp (M.F.); sokamoto@sahs.med.osaka-u.ac.jp (S.O.); 2Department of Clinical Laboratory, Kindai University Hospital, Osakasayama, Osaka 589-8511, Japan; 3Department of Infectious Diseases, Tokyo Women’s Medical University, Shinjuku-ku, Tokyo 162-8666, Japan; arai.yuko@twmu.ac.jp (Y.A.); uzawa.yutaka@twmu.ac.jp (Y.U.); 4Department of Clinical Laboratory Medicine, Faculty of Medicine, Kindai University, Osakasayama, Osaka 589-8511, Japan; kamisako@med.kindai.ac.jp; 5Division of Food Science and Biotechnology, Graduate School of Agriculture, Kyoto University, Uji, Kyoto 611-0011, Japan; ogura.kohei.7x@kyoto-u.ac.jp

**Keywords:** antimicrobial susceptibility, *Gemella bergeri*, *Gemella haemolysans* group, *Gemella morbillorum*, *Gemella taiwanensis*, *Gemella sanguinis*, *gyrA*, macrolide resistance, quinolone resistance, tetracycline resistance, quinolone resistance-determining region

## Abstract

*Gemella* is a catalase-negative, facultative anaerobic, Gram-positive coccus that is commensal in humans but can become opportunistic and cause severe infectious diseases, such as infective endocarditis. Few studies have tested the antimicrobial susceptibility of *Gemella*. We tested its antimicrobial susceptibility to 27 drugs and defined the resistant genes using PCR in 58 *Gemella* strains, including 52 clinical isolates and 6 type strains. The type strains and clinical isolates comprised 22 *G. morbillorum*, 18 *G. haemolysans* (GH) group (genetically indistinguishable from *G. haemolysans* and *G. parahaemolysans*), 13 *G. taiwanensis*, three *G. sanguinis*, and two *G. bergeri*. No strain was resistant to beta-lactams and vancomycin. In total, 6/22 (27.3%) *G. morbillorum* strains were erythromycin- and clindamycin-resistant *ermB*-positive, whereas 5/18 (27.8%) in the GH group, 6/13 (46.2%) *G. taiwanensis*, and 1/3 (33.3%) of the *G. sanguinis* strains were erythromycin-non-susceptible *mefE*- or *mefA*-positive and clindamycin-susceptible. The MIC_90_ of minocycline and the ratios of *tetM*-positive strains varied across the different species—*G. morbillorum*: 2 µg/mL and 27.3% (6/22); GH group: 8 µg/mL and 22.2% (4/18); *G. taiwanensis*: 8 µg/mL and 53.8% (7/13), respectively. Levofloxacin resistance was significantly higher in *G. taiwanensis* (8/13, 61.5%) than in *G. morbillorum* (2/22, 9.1%). Levofloxacin resistance was associated with a substitution at serine 83 for leucine, phenylalanine, or tyrosine in GyrA. The mechanisms of resistance to erythromycin and clindamycin differed across *Gemella* species. In addition, the rate of susceptibility to levofloxacin differed across *Gemella* spp., and the quinolone resistance mechanism was caused by mutations in GyrA alone.

## 1. Introduction

*Gemella* is a catalase-negative, facultative anaerobic coccus [1]. *Gemella* is Gram-positive; however, its stain can be easily decolorized [2]. Cells appear as single cells, pairs, tetrads, and sometimes irregular clusters. As of July 2023, 10 *Gemella* sp. have been identified: *G. asaccharolytica* [3], *G. bergeri* [4], *G. cuniculi* [5], *G. haemolysans* [6], *G. morbillorum* [7], *G. palaticanis* [8], *G. parahaemolysans* [2], *G. sanguinis* [9], *G. taiwanensis* [2], and *G. massiliensis* [10,11]. *G. massiliensis* was recently categorized as a new species according to the International Code of Nomenclature of Prokaryotes. These *Gemella* species are commensals in the oral cavities and guts of humans and are occasionally isolated as pathogens from patients with infective endocarditis [12,13,14,15,16,17,18], cerebral abscesses [19], bacteremia [20,21,22], septic shock [23], meningitis [24], and purpura fulminans with Lemierre’s syndrome [25], spondylodiscitis [26]. Phenotypically, *Gemella* resembles viridans group streptococci, *Abiotrophia*, or *Granulicatella* and has thus been misdiagnosed and underestimated in clinical microbiology laboratories [27]. Some *Gemella* sp., such as *G. haemolysans*, *G. parahaemolysans*, and *G. taiwanensis*, cannot be distinguished even using matrix-assisted laser desorption ionization–time-of-flight mass spectrometry [12] or 16S rDNA sequencing [2,12,28].

The global spread of antimicrobial resistance in pathogenic bacteria is a growing concern. Some *Gemella* sp. are reportedly resistant to macrolides [2,29,30] and tetracyclines [29,30]. Studies have shown that in *Gemella* sp., *mefA*, *mefE*, and *ermB* are involved in macrolide resistance and *tetO* and *tetM* are involved in tetracycline resistance [2,29,30,31]. In 2016, document M45-Third Edition of the Clinical and Laboratory Standards Institute (CLSI) described a standardized method for antimicrobial resistance testing for *Gemella* sp. [32]; however, few studies have used this method [22,25,33]. In many cases, clinicians and laboratory technicians are using methods and setting breakpoints without identical criteria. Additionally, it remains unknown whether *Gemella* sp. has resistance mechanisms for other antimicrobials, such as quinolones.

In this study, we tested the antimicrobial susceptibility of 52 clinical isolates and six type strains of *Gemella* according to CLSI M45-Third Edition [32]. Additionally, we analyzed macrolide-, tetracycline-, and quinolone-resistant genes in the *Gemella* strains.

## 2. Results

### 2.1. Identification Using 16S rDNA Sequencing and Multilocus Sequence Analysis

Of the 52 clinical isolates of *Gemella*, strains 21, 2, and 1 were considered *G. morbillorum*, *G. sanguinis*, and *G. bergeri*, respectively, using 16S rDNA sequencing. The origins of the sources are presented in Table 1. The other 28 isolates were assigned to *G. haemolysans*, *G. parahaemolysans*, and *G. taiwanensis.* Discriminating the three species based on 16S rDNA sequencing alone was challenging, owing to the high homology between the species. Therefore, we conducted multilocus sequence analysis (MLSA) [2] using *groEL*, *recA*, and *rpoB* sequences. In total, isolates 4, 3, and 12 were assigned to *G. haemolysans*, *G. parahaemolysans*, and *G. taiwanensis*, respectively, using both 16S rDNA sequencing and MLSA. Consistently, the 12 isolates typed as *G. taiwanensis* were identified as *G. taiwanensis* using MLSA with high homology. However, the 16 isolates belonging to *G. haemolysans* or *G. parahaemolysans* could not be distinguished even when using MLSA. These were defined as the *G. haemolysans*—*parahaemolysans* (GH) group (Appendix A). The MLSA homology of strains TWCC 53044 and TWCC 58771 to the type strains of *Gemella* is divergent, suggesting that they are new species of *Gemella*. In this study, these strains were tentatively assigned to the GH group and *G. taiwanensis*, respectively.

### 2.2. Antimicrobial Susceptibility

#### 2.2.1. Susceptibility to Penicillin G, Cefotaxime, Ceftriaxone, Meropenem, and Vancomycin

The clinical isolates and five type strains (*G. morbillorum* ATCC 27824^T^, *G. haemolysans* ATCC 10379^T^, *G. parahaemolysans* JCM 18067^T^, *G. taiwanensis* JCM 18066^T^, and *G. sanguinis* CCUG 37820^T^) were susceptible to penicillin G. Because most of the Gemella strains used in our study were highly sensitive to the β-lactams, the MIC_50_ and MIC_90_ of the drugs became close or the same. However, the *G. taiwanensis* type strain JCM 18066^T^ had intermediate susceptibility. The MIC_90_ value of the 58 *Gemella* strains was <0.06 µg/mL. All the isolates and type strains were susceptible to ceftriaxone (MIC_90_ ≤ 0.06 µg/mL), cefotaxime (MIC_90_ = 0.12 µg/mL), meropenem (MIC_90_ ≤ 0.06 µg/mL), and vancomycin (MIC_90_ = 0.50 µg/mL) (Table 2). Because three strains of *G. morbillorum* (TWCC 57201, TWCC 57818, and TWCC 71529) grew slower than other strains, the MIC of each drug was determined at 72–96 h (Appendix A, Figure 1).

#### 2.2.2. Susceptibility to Erythromycin

In total, 20/58 strains were erythromycin-non-susceptible (intermediate or resistant), with MIC_90_ > 2 µg/mL. Although the ratios of the erythromycin-non-susceptible isolates varied across species, there was no significant difference among *G. morbillorum*, the GH group, and *G. taiwanensis*—*G. morbillorum*: 27.3% (6/22), GH group: 38.9% (7/18), *G. taiwanensis*: 46.2% (6/13), *G. sanguinis*: 33.3% (1/3), and *G. bergeri*: 0.0% (0/2) (Table 2, Figure 1 and Figure 2).

#### 2.2.3. Susceptibility to Clindamycin

In total, 10/58 strains were clindamycin-non-susceptible, resulting in MIC_90_ > 2 µg/mL. Clindamycin-resistant *G. taiwanensis*, *G. sanguinis*, and *G. bergeri* isolates were not detected, and differences were not significant—*G. morbillorum*: 27.3% (6/22), GH group: 22.2% (4/18), *G. taiwanensis*: 0.0% (0/13), and *G. sanguinis*: 0.0% (0/3) (Table 2, Figure 1 and Figure 2). Interestingly, all six erythromycin-resistant *G. morbillorum* strains were clindamycin-resistant. In contrast, 5/7 GH group strains, six strains of *G. taiwanensis*, and one *G. sanguinis* strain were erythromycin-non-susceptible and clindamycin-susceptible.

#### 2.2.4. Susceptibility to Levofloxacin

In total, 21/58 strains were levofloxacin-resistant, resulting in MIC_90_ > 128 µg/mL. Ratios of the levofloxacin strains varied across species—*G. morbillorum*: 9.1% (2/22), GH group: 50.0% (9/18), *G. taiwanensis*: 61.5% (8/13), *G. sanguinis*: 66.7% (2/3), and *G. bergeri*: 0.0% (0/2). The ratio of the resistant strains was significantly higher in *G. taiwanensis* than in *G. morbillorum* (*p* < 0.05 using chi-squared test), whereas the ratio was higher in the GH group than in *G. morbillorum* (*p* = 0.08) (Table 2, Figure 1).

#### 2.2.5. Susceptibility to Minocycline

Although the MIC_90_ value of the tetracycline antibiotic minocycline was 8 μg/mL, overall, the values were lower in *G. morbillorum* (2 μg/mL) than in the GH group (8 μg/mL) and G. *taiwanensis* (8 μg/mL) (Table 2).

#### 2.2.6. Susceptibility to Other Antimicrobial Agents

We tested the 18 antimicrobial agents whose breakpoints are not listed in CLSI M45-third edition. *Gemella* strains showed low MIC values for all beta-lactams: ampicillin, amoxicillin–clavulanic acid, sulbactam–ampicillin, cefazolin, cefdinir, cefepime, and imipenem (MIC_90_: ≤0.12, ≤0.25/0.12, ≤0.06/0.12, ≤0.25, ≤0.25, ≤0.06, and ≤0.06 μg/mL, respectively). The MIC_90_ values of clarithromycin and azithromycin were 8 and >4 µg/mL, respectively, consistent with those of erythromycin. The MIC_90_ values of clarithromycin varied among *G. morbillorum* (>16 μg/mL), the GH group (8 μg/mL), and *G. taiwanensis* (2 μg/mL), indicating the acquisition of high resistance to clarithromycin in *G. morbillorum* strains. The MIC_90_ value of moxifloxacin was high (>2 μg/mL) in *Gemella* strains. The MIC_90_ values of the aminoglycoside antibiotics gentamicin, gentamicin500 (to confirm tolerance to high concentrations of gentamicin), and arbekacin were 8, ≤500, and >8 μg/mL, respectively; sulfamethoxazole–trimethoprim, fosfomycin, and rifampicin were >38/2, ≤16, and ≤0.5 μg/mL, respectively; and the anti-MRSA agents teicoplanin, linezolid, and daptomycin were ≤0.5, 1, and 2 μg/mL, respectively (Table 2). Typically, streptococci are aminoglycoside-resistant. Therefore, we tested gentamicin500 to identify any *Gemella* strains that are highly resistant to aminoglycoside.

### 2.3. Phenotypes and Genotypes of Macrolide-Resistant Strains

The six erythromycin–clindamycin-resistant *G. morbillorum* strains exhibited constitutive resistance to macrolide, lincosamide, and streptogramin B (cMLSB). Their genotypes—*mefA/E*-negative, *ermB-*positive, and *msrA*-negative—were consistent with their phenotypes. Furthermore, five/seven strains of the GH group, six strains of *G. taiwanensis*, and one strain of *G. sanguinis* which were erythromycin-non-susceptible and clindamycin-susceptible, had macrolide-resistant (M) phenotypes and *mefE*- (four strains) or *mefA*-positive (one strain), *erm*-negative, and *msrA*-negative genotypes. In total, 2/7 GH group strains (TWCC 59567 and TWCC 59795) were erythromycin-resistant and clindamycin-non-susceptible and *mefE*-positive, but showed M phenotype. These results show that erythromycin-resistant *G. morbillorum* is associated with *ermB*, and erythromycin-non-susceptible GH-group, *G. taiwanensis*, and *G. sanguinis* are associated with *mefE*. The MIC values for clarithromycin were higher in the six *ermB*-positive *G. morbillorum* strains (8 or >16 µg/mL) (Table 3). All erythromycin-susceptible *Gemella* strains, except the *G. sanguinis* strain TWCC 70419, lacked *mefA/E*, *erm*, or *msrA* (Appendix A).

### 2.4. Tetracycline Resistance

Next, we analyzed the possession rates of *tet*. Overall, 17/58 (29.3%) strains were *tetM*-positive; none of the other *tet* genes was detected. The ratios of *tetM*-positive strains in *G. morbillorum*, the GH group, *G. taiwanensis*, *G. sanguinis*, and *G. bergeri* were 27.3% (6/22), 33.3% (6/18), 38.5% (5/13), 0/3 (0.0%), and 0.0% (0/2), respectively. Among the 41 *tetM*-negative strains, one had minocycline (MIC = 2 µg/mL). The minocycline MIC values of the others were ≤1 µg/mL. The minocycline MIC of the 17 *tetM*-positive strains varied: ≤1 for five, 2 for five, and ≥8 µg/mL for seven strains, respectively (Table 4).

### 2.5. Mutations in gyrA and gyrB

We analyzed the *gyrA* and *gyrB* sequences. The 35 quinolone-susceptible strains possessed *gyrA*, encoding GyrA with a serine residue at 83 (S83). The serine residue was substituted with leucine (S83L), phenylalanine (S83F), or tyrosine (S83Y) in the 21 quinolone-resistant strains. Specifically, two *G. morbillorum* strains possessed GyrA/S83L, encoding *gyrA*. Seven of the GH group, seven *G. taiwanensis*, and two *G. sanguinis* strains contained S83F. Two in the GH group and one *G. taiwanensis* strains contained S84Y. GyrB mutations associated with levofloxacin resistance were not detected (Table 5).

## 3. Discussion

In this study, we tested the antimicrobial susceptibility of 52 clinical isolates and six type strains of *Gemella* sp. with 27 drugs in accordance with CLSI M45-Third Edition [32]. Discriminating between *G. haemolysans* and *G. parahaemolysans* was difficult using MLSA. Therefore, the strains that could not be differentiated were assigned to the GH group. Garcia Lopez et al. [34] proposed grouping the four strains registered as *G. haemolysans* as the “Haemolysans group” because their average nucleotide identity ranged from 87.2% to 99.9%.

Although susceptibility was judged after 48 h of incubation for most cases, some strains needed 72–96 h incubation. Some studies have used the E-test, using optimal culture media for *Gemella*, because bacterial growth is poor with the CLSI method [26,35]. To ensure accurate tests for antimicrobial susceptibility, it might be important to update culture conditions, such as adding supplemental nutrition, to promote better growth of *Gemella* strains.

All the strains were susceptible to beta-lactams, except the *G. taiwanensis* type strain JCM 18066^T^, which had intermediate susceptibility to penicillin G. Because all the *Gemella* strains used in our study were highly sensitive to β-lactams, the MIC_50_ and MIC_90_ values of the drugs were similar or the same. *Gemella* is usually susceptible to beta-lactams [2,14,20,21,22,23,25,35,36]; however, there are some reports of resistance to penicillin G [2,21,23,24], ceftriaxone [24], and meropenem [33]. Overall susceptibility rates for erythromycin, clindamycin, and levofloxacin were 65.5%, 82.8%, and 63.8%, respectively. Consistently, Baghdadi et al. [33] reported that the susceptibility rates of 14 strains of *Gemella* (not speciated) were 50% for erythromycin, 86% for clindamycin, and 50% for levofloxacin. For *G. morbillorum*, our MIC_90_ value (>2 μg/mL) for clindamycin was different from that reported in another study (MIC_90_ ≤ 0.06 μg/mL) [36], indicating that trends in antimicrobial susceptibility vary among reports. Therefore, antimicrobial susceptibility must be tested for all *Gemella* isolates, especially those isolated from sterile sites, such as blood, because the isolate is suspected to be a pathogen. In this study, *G. morbillorum* strains, including type strains, were frequently isolated from sterile materials, such as blood and ascites, as well as wounds (Table 1 and Appendix A). In contrast, the GH group and *G. taiwanensis* strains were derived from respiratory tissues, such as the pharynx (Table 1 and Appendix A). This suggests that the pathogenicity and usual colonization sites of *Gemella* differ across species.

Drugs with no breakpoints in the CLSI M45-Third Edition had the same trend as the beta-lactams, macrolides, and quinolones of the same family. Rifampicin and anti-MRSA drugs have low MIC_90_ values and may be therapeutic options.

Resistance of streptococci to MLS_B_ antibiotics occurs through two major mechanisms. The first is mediated by the methylation of ribosomal targets of these antibiotics (MLS_B_ resistance). The methylase responsible for this activity is encoded by *erm*. MLS_B_ resistance can be constitutive (cMLS_B_) or inducible (iMLS_B_). MLS_B_-mediated resistance by *erm* confers strong resistance to MLS_B_ [37]. The second involves an active efflux system associated with *mef*, which exhibits low resistance to 14- and 15-membered macrolides only, and the resulting phenotype is M [38,39].

We found that the erythromycin-resistant *G. morbillorum* possessed *ermB*, whereas the erythromycin-resistant GH group, *G. taiwanensis*, and *G. sanguinis* had *mefE*. Consistent with our data, reports show that the MIC values of erythromycin are 2 [2] and 1 μg/mL [31] for *mef*-positive *G. haemolysans* and *G. taiwanensis*, respectively. The *G. haemolysans* strain possesses *mef* [2], and the *G. taiwanensis* strain possesses *mef* but not *ermT*, *ermTR*, or *ermB* [31]. Conversely, Zolezzi et al. [29] detected *G. morbillorum* with *mefA/E*, *G. haemolysans* with phenotype cMLS_B_ and *ermB*, and *G. morbillorum* with iMLS_B_-resistant phenotype and *ermB*. Although the relationship between gene acquisition and *Gemella* sp. Is unknown, *Gemella* sp. *mefE* shares 99%–100% homology with *Streptococcus pneumoniae* (European Nucleotide Archive (ENA) Accession No. U83667.1) and *Streptococcus salivarius* (ENA Accession No. CAC87432.1) *mefE*, suggesting a genetic exchange between streptococci. In our collection, the two erythromycin-resistant GH group strains (TWCC 59567 and TWCC 59795) were categorized as *mefE*-positive clindamycin-resistant and intermediate, respectively. Although one *G. sanguinis* strain (TWCC 70419) possessed *mefE*, it was susceptible to erythromycin, with a low MIC value of clarithromycin (≤0.12 μg/mL) and contained no mutation in *mefE* (Appendix A). The MIC of azithromycin for the *G. sanguinis* strain was relatively high (0.25 μg/mL), indicating that *mefE* is partially involved in susceptibility to azithromycin. The *ermB*-positive *G. morbillorum* strains showed higher MIC values to erythromycin, clarithromycin, and azithromycin than to the *mefE*-positive GH group, *G. taiwanensis*, *G. sanguinis*, and *G. bergeri* strains (Table 3). Our results suggest that *Gemella* sp. with *erm* possess higher macrolide resistance than those harboring *mef*. Consistently, macrolide resistance was higher after the acquisition of *erm* than that of *mef* [38,39,40]. Although we did not find *msrA*-positive strains in our collection, Zolezzi et al. reported *msrA* + *G. morbillorum* [30]. Further analysis must clarify the acquisition of macrolide resistance by *Gemella* sp.

Because *ermB*, *mefE*, and *tetM* are common to viridans group streptococci, etc., it is assumed that there was horizontal gene transfer between them. Zolezzi et al. performed in vitro *mefE* transfer from *Gemella* sp. and viridans group streptococci to *S. pneumoniae* [29]. Streptococcal *ermB* and *tetM* are associated with Tn916- and/or Tn916-like conjugative transposons. In this study, *ermB*, *mefE*, and *tetM* of *Gemella* sp. showed high homology with those of *S. pneumoniae*, indicating gene transfer from *S. pneumoniae* to *Gemella* sp. via the Tn916 family. In total, 12/13 (92.3%) *Gemella* strains with a minocycline MIC value ≥ 2 μg/mL harbored *tetM*. Although our data showed possession of only *tetM*, Zolezzi et al. reported that *G. morbillorum* and *G. haemolysans* possess both *tetM* and *tetO* [30]. The oral cavity is a suitable environment for horizontal gene transfer because commensal bacteria exist in close proximity to plaques [41]. In a systematic review, Brooks et al. concluded that *tetM* and Tn916 were the most prevalent gene and mobile genetic element associated with antibiotic resistance in the oral cavity, respectively, and the most common resistance genes varied in these sites, such as *tetM* in the root canal and *ermB* in supragingival plaques [42]. Rossi-Fedele et al. reported that Tn916 is involved in the transfer of *tetM* from *Neisseria niger* to *Enterococcus faecalis* in the root canal [43]. Villedieu et al. showed that *tetM* and *ermB* could transfer to *Enterococcus faecalis* through the Tn916-like conjugative transposon Tn1545 [44]. Zolezzi et al. performed in vitro *mefE* gene transfer from *Gemella* species and viridans group streptococci to *S. pneumoniae* [29].

There are three quinolone resistance mechanisms. The first involves reduced drug binding to the enzyme–DNA complex due to resistance mutations in one or both quinolone target enzymes, DNA gyrase and DNA topoisomerase IV. The second involves a resistance mutation in a regulatory gene that controls the expression of the native efflux pump in the bacterial membrane. The third involves a resistance gene acquired on plasmids [45]. In this study, we focused on mutations in DNA gyrase. Quinolones target two essential bacterial type II topoisomerase enzymes—DNA gyrase and DNA topoisomerase IV. Each enzyme is a heterotetramer: gyrase contains two GyrA and two GyrB subunits, whereas topoisomerase IV contains two ParC and two ParE subunits. GyrA is homologous to ParC, and GyrB is homologous to ParE [45]. In Gram-positive bacteria, the *gyrA* mutation follows the *parC* mutation. Quinolone resistance in viridans group streptococci [46] and β-hemolytic *Streptococcus* spp. [47] is higher for *parC* + *gyrA* mutations than for *parC* mutations. Mutations in streptococcal *gyrA* alone show high resistance to quinolones [47].

*Gemella* lacks topoisomerase IV, indicating that the Ser83 mutation in GyrA is only responsible for high resistance to levofloxacin. Similar results were observed for *Helicobacter pylori* [48] and *Mycobacterium tuberculosis* [49], which lack topoisomerase IV; however, their quinolone resistance was attributed to mutations in the quinolone resistance-determining regions of *gyrA*. In Japan, the quinolone resistance rates of *Gemella* tended to be higher than those for similar *Abiotrophia* and *Granulicatella* sp. [50], as well as *S. pneumoniae* [51]. Because *Gemella* lacks *parC*, a single mutation in *gyrA* can occur easily, resulting in the acquisition of higher quinolone resistance than these streptococci. The consumption of oral quinolones is higher in Japan than in other countries, suggesting that high quinolone exposure [52] resulted in the frequent emergence of quinolone-resistant *Gemella*. The National Action Plan on Antimicrobial Resistance of Japan recommended that oral prescription of quinolones must be reduced for control of quinolone-resistance.

This study has some limitations. We did not collect patient information, such as clinical history, antibiotics used for treatment, prognosis, isolated hospitals, and time period of collection. To further analyze the characteristics of *Gemella* species, patient information might be helpful.

In conclusion, the mechanisms of macrolide resistance and occupancy of the levofloxacin resistance gene (*gyrA*) varied across *Gemella* sp. Our data suggest that species-level identification is required for further characterization of antimicrobial resistance.

## 4. Materials and Methods

### 4.1. Bacterial Strains

We collected 52 *Gemella* strains isolated from clinical sites, such as blood, wound pus, abdominal drain effluent, ascites, closed wounds, open wounds, catheter urine, sputum, lung biopsy needle wash solution, corneal abrasion, bile, and noses (Appendix A and Table 1). Five type strains, namely *G. morbillorum* ATCC 27824^T^, and *G. haemolysans* ATCC 10379^T^ were obtained from American Type Culture Collection Manassas, VA, USA, *G. parahaemolysans* JCM 18067^T^, and *G. taiwanensis* JCM 18066^T^, were collected from the Japan Collection of Microorganisms, RIKEN BRC, Ibaraki, Japan. *G. sanguinis* CCUG 37820^T^ and *G. bergeri* CCUG 37817^T^ were obtained from the Culture Collection University of Götheborg, Götheborg, Sweden. All the strains were stored in 10% skim milk at −80 °C until use.

### 4.2. DNA Extraction

Bacteria grown on 5% sheep blood agar EX plates (Nissui Pharmaceutical Co., Ltd., Tokyo, Japan) were suspended in McFarland 2.0 standard in 100 μL of 10 mM Tris-HCl and 1 mM EDTA (TE) buffer (pH 8.0) supplemented with 320 U of achromopeptidase (Wako Chemical, Osaka, Japan), followed by incubation at 55 °C for 15 min. After centrifugation at 15,000× *g* for 5 min, the supernatant was used as a crude DNA template for PCR.

### 4.3. Identification of Gemella sp.

*Gemella* species were identified based on 16S rDNA sequencing and MLSA [2], which utilize concatenated sequences of *groEL*, *recA*, and *rpoB*. Primers for MLSA were designed by Hung et al. [28] and our team. The primers used are listed in Appendix A. The genes were amplified using TaKaRa Ex Taq polymerase Hot Start Version (TaKaRa Bio, Shiga, Japan) according to the following thermal cycle: initial denaturation at 95 °C for 10 min, 30 cycles of denaturation at 95 °C for 30 s, primer annealing at the indicated temperatures (Appendix A) for 1 min, and extension at 72 °C for 1–4 min depending on product size (1 kb/min), and final extension at 72 °C for 5 min. The amplicons were purified and applied to dye terminator cycle sequencing using each of the primers and the BigDye Terminator v3.1 Cycle Sequencing Kit (Thermo Fisher Scientific, Waltham, MA, USA) according to the manufacturers’ instructions. After clean-up using the BigDye XTerminator Purification Kit (Thermo Fisher Scientific), the sequences were analyzed using an automatic DNA sequencer (ABI Prism 310 genetic analyzer; Applied Biosystems, Foster City, CA, USA).

### 4.4. Detection of Macrolide-Resistant Genes

Macrolide resistance-related genes *mefA/E* [30], *ermA* [53], *ermB* [54], *ermC* [53], *ermM* [53], *ermTR* [53], and *msrA* [53] were detected according to published reports. Sanger sequencing was conducted to confirm *mefA* or *mefE*. Amplified PCR fragments were analyzed using sequencing, as described in Section 4.3.

### 4.5. Detection of Tetracycline-Resistant Genes

Tetracycline-resistant genes *tetM*, *tetO, tetK*, *tetL*, *tetT*, *tetS*, and *tetW* were detected as described, with slight modifications [53,55]. Briefly, the genes were amplified using the TaKaRa Ex Premier DNA Polymerase (TaKaRa Bio) and pairs of primers (Appendix A) with the following thermal cycle: initial denaturation at 94 °C for 5 min, 30 cycles of denaturation at 98 °C for 10 s, primer annealing at the indicated temperatures (Appendix A) for 15 s, and extension at 68 °C for 70 s, and final extension at 72 °C for 5 min.

### 4.6. Detection of Quinolone-Resistant Genes gyrA and gyrB

Sequences of *gyrA* and *gyrB* were amplified with our designed primers (Appendix A) and analyzed using sequencing. Gene sequences were translated into amino acid sequences, followed by alignment and detection of amino acid substitutions.

### 4.7. Antimicrobial Susceptibility Test

The antimicrobial susceptibility test was conducted in accordance with the CLSI M45-Third Edition for penicillin G, ampicillin, amoxicillin/clavulanic acid, sulbactam/ampicillin, cefazolin, cefdinir, ceftriaxone, cefotaxime, cefepime, imipenem, meropenem, erythromycin, clarithromycin, azithromycin, clindamycin, erythromycin/clindamycin, levofloxacin, moxifloxacin, minocycline, sulfamethoxazole/trimethoprim, gentamicin, gentamicin500, arbekacin, fosfomycin, rifampicin, vancomycin, teicoplanin, linezolid, and daptomycin [32]. Briefly, *Gemella* strains grown on 5% sheep blood agar EX plates (Shimadzu Diagnostics Corporation, Tokyo, Japan) were suspended in saline and inoculated in Difco cation-adjusted Mueller Hinton Broth (Becton, Dickinson, Sparks, MD, USA) supplemented with lysed 5% horse blood (KOHJIN BIO, Saitama, Japan) at 5 × 10^5^ CFU/mL. Dry plates 34, 42, and 44 were purchased from Eiken Chemical Co., Ltd. (Tokyo, Japan). Dry plates were incubated at 35 °C and 5% CO_2_ for 48 h. In cases of poor growth until 48 h, the strains were cultured for 72–96 h. MICs were interpreted using the CLSI M45 breakpoints for *Gemella* spp. when available [32]. For quality control, CLSI-Third Edition-recommended *Streptococcus pneumoniae* ATCC 49619 was used [32]. Quality control was performed for each change of plates in every lot.

### 4.8. Statistical Analysis

Ratios of resistant and non-susceptible strains were statistically analyzed using the Chi-square test.

### 4.9. Ethical Approval

The isolation, storage, and utilization of clinical strains were conducted according to the guidelines of each of the participating hospitals.

## Figures and Tables

**Figure 1 antibiotics-12-01538-f001:**
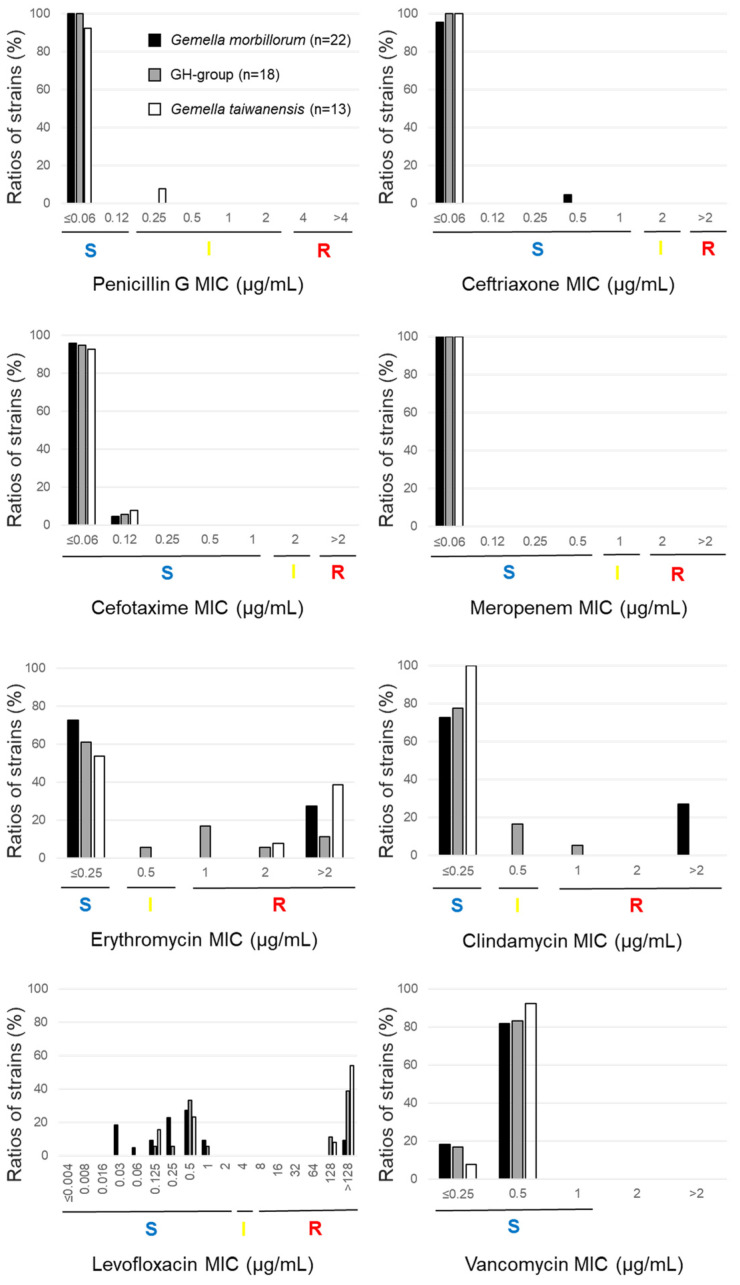
Ratios of resistant strains. S (blue), I (yellow), and R (Red) indicate sensitive, intermediate, and resistant, respectively.

**Figure 2 antibiotics-12-01538-f002:**
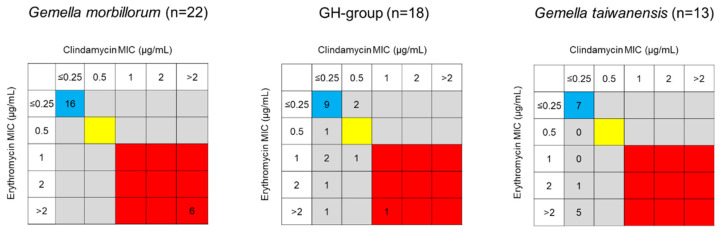
Distribution of erythromycin/clindamycin resistance in *Gemella* strains. Blue, yellow, and red boxes indicate sensitive, intermediate, and resistant, respectively.

**Table 1 antibiotics-12-01538-t001:** Isolated sites of *Gemella* species used in this study.

Specimen	Number of Strains				
	*Gemella morbillorum*	GHgroup	*Gemella taiwanensis*	*Gemella sanguinis*	*Gemella bergeri*
Blood	8	4	5	1	
Ascites	2				
Bile		1			
Pleural effusion				1	
Wound pus	10	2	4		1
Sputum		2			
Lung biopsy		1			
Pharynx		5	2		
Nose			1		
Urine	1				
Cornea		1			
Total	21	16	12	2	1

Isolated sites of *Gemella type* strain. *G. morbillorum* ATCC 27824^T^: lung abscess, *G. haemolysans* ATCC 10379^T^: unknown, *G. parahaemolysans* JCM 18067^T^: blood, *G. taiwanensis* JCM 18066^T^: blood, *G. sanguinis* CCUG 37820^T^: bood, *G. bergeri* CCUG 37817^T^: blood.

**Table 2 antibiotics-12-01538-t002:** Susceptibility to antimicrobial agents with breakpoints listed in CLSI M45-third edition.

Antimicrobial Agents/ *Gemella* spp.	MIC (μg/mL)			Interpretive Breakpoint (μg/mL) ^a^ or % of Isolates		
	Range	MIC_50_	MIC_90_	Susceptible	Intermediate	Resistant
Penicillin G ^c^	≤0.06–>4			≤0.12	0.25–2	≥4
*Gemella morbillorum*	≤0.06	≤0.06	≤0.06	100.0	0.0	0.0
GH group	≤0.06	≤0.06	≤0.06	100.0	0.0	0.0
*Gemella taiwanensis*	≤0.06–0.25	≤0.06	≤0.06	92.3	7.7	0.0
*Gemella sanguinis*	≤0.06	–	–	100.0	0.0	0.0
*Gemella bergeri*	≤0.06	–	–	100.0	0.0	0.0
Total	≤0.06–0.25	≤0.06	≤0.06	98.3	1.7	0.0
Ampicillin	≤0.12–>4					
*Gemella morbillorum*	≤0.12–0.25	≤0.12	≤0.12	NA ^b^	NA	NA
GH group	≤0.12	≤0.12	≤0.12	NA	NA	NA
*Gemella taiwanensis*	≤0.12–0.5	≤0.12	≤0.12	NA	NA	NA
*Gemella sanguinis*	≤0.12	–	–	NA	NA	NA
*Gemella bergeri*	≤0.12	–	–	NA	NA	NA
Total	≤0.12–0.5	≤0.12	≤0.12	NA	NA	NA
Amoxicillin–clavulanic acid	≤0.25/0.12–>4/2					
*Gemella morbillorum*	≤0.25/0.12	≤0.25/0.12	≤0.25/0.12	NA	NA	NA
GH group	≤0.25/0.12	≤0.25/0.12	≤0.25/0.12	NA	NA	NA
*Gemella taiwanensis*	≤0.25/0.12	≤0.25/0.12	≤0.25/0.12	NA	NA	NA
*Gemella sanguinis*	≤0.25/0.12	–	–	NA	NA	NA
*Gemella bergeri*	≤0.25/0.12	–	–	NA	NA	NA
Total	≤0.25/0.12	≤0.25/0.12	≤0.25/0.12	NA	NA	NA
Sulbactam–ampicillin	≤0.06/0.12–>2/4					
*Gemella morbillorum*	≤0.06/0.12	≤0.06/0.12	≤0.06/0.12	NA	NA	NA
GH group	≤0.06/0.12	≤0.06/0.12	≤0.06/0.12	NA	NA	NA
*Gemella taiwanensis*	≤0.06/0.12–0.25/0.5	≤0.06/0.12	≤0.06/0.12	NA	NA	NA
*Gemella sanguinis*	≤0.06/0.12	–	–	NA	NA	NA
*Gemella bergeri*	≤0.06/0.12	–	–	NA	NA	NA
Total	≤0.06/0.12–0.25/0.5	≤0.06/0.12	≤0.06/0.12	NA	NA	NA
Cefazolin	≤0.25–>2					
*Gemella morbillorum*	≤0.25	≤0.25	≤0.25	NA	NA	NA
GH group	≤0.25–0.5	≤0.25	0.5	NA	NA	NA
*G* *emellataiwanensis*	≤0.25–0.5	≤0.25	0.5	NA	NA	NA
*Gemella sanguinis*	≤0.25	–	–	NA	NA	NA
*Gemella bergeri*	≤0.25	–	–	NA	NA	NA
Total	≤0.25–0.5	≤0.25	≤0.25	NA	NA	NA
Cefdinir	≤0.25–>1					
*G* *emella morbillorum*	≤0.25	≤0.25	≤0.25	NA	NA	NA
GH group	≤0.25	≤0.25	≤0.25	NA	NA	NA
*Gemella taiwanensis*	≤0.25	≤0.25	≤0.25	NA	NA	NA
*Gememlla sanguinis*	≤0.25–0.5	–	–	NA	NA	NA
*Gemella bergeri*	≤0.25	–	–	NA	NA	NA
Total	≤0.25–0.5	≤0.25	≤0.25	NA	NA	NA
Ceftriaxone ^c^	≤0.06–>2			≤1	2	≥4
*Gemella morbillorum*	≤0.06–0.5	≤0.06	≤0.06	100.0	0.0	0.0
GH group	≤0.06	≤0.06	≤0.06	100.0	0.0	0.0
*Gemella taiwanensis*	≤0.06	≤0.06	≤0.06	100.0	0.0	0.0
*Gemella sanguinis*	0.25–1	–	–	100.0	0.0	0.0
*Gemella bergeri*	≤0.06			100.0	0.0	0.0
Total	≤0.06–1	≤0.06	≤0.06	100.0	0.0	0.0
Cefotaxime ^c^	≤0.06–>2			≤1	2	≥4
*Gemella morbillorum*	≤0.06–0.12	≤0.06	≤0.06	100.0	0.0	0.0
GH group	≤0.06–0.12	≤0.06	≤0.06	100.0	0.0	0.0
*Gemella taiwanensis*	≤0.06–0.12	≤0.06	≤0.06	100.0	0.0	0.0
*Gemella sanguinis*	0.25–1	–	–	100.0	0.0	0.0
*Gemella bergeri*	≤0.06	–	–	100.0	0.0	0.0
Total	≤0.06–1	≤0.06	0.12	100.0	0.0	0.0
Cefepime	≤0.06–>2					
*G* *emella morbillorum*	≤0.06–0.5	≤0.06	≤0.06	NA	NA	NA
GH group	≤0.06–0.12	≤0.06	0.12	NA	NA	NA
*Gemella taiwanensis*	≤0.06–0.12	≤0.06	≤0.06	NA	NA	NA
*Gemella sanguinis*	0.25–1	–	–	NA	NA	NA
*Gemella bergeri*	≤0.06	–	–	NA	NA	NA
Total	≤0.06–1	≤0.06	0.12	NA	NA	NA
Imipenem	≤0.06–>4					
*Gemella morbillorum*	≤0.06	≤0.06	≤0.06	NA	NA	NA
GH group	≤0.06	≤0.06	≤0.06	NA	NA	NA
*Gemella taiwanensis*	≤0.06	≤0.06	≤0.06	NA	NA	NA
*Gemella sanguinis*	≤0.06	–	–	NA	NA	NA
*Gemella bergeri*	≤0.06	–	–	NA	NA	NA
Total	≤0.06	≤0.06	≤0.06	NA	NA	NA
Meropenem ^c^	≤0.06–>2			≤0.5	1	≥2
*Gemella morbillorum*	≤0.06	≤0.06	≤0.06	100.0	0.0	0.0
GH group	≤0.06	≤0.06	≤0.06	100.0	0.0	0.0
*Gemella taiwanensis*	≤0.06	≤0.06	≤0.06	100.0	0.0	0.0
*Gemella sanguinis*	≤0.06	–	–	100.0	0.0	0.0
*Gemella bergeri*	≤0.06	–	–	100.0	0.0	0.0
Total	≤0.06	≤0.06	≤0.06	100.0	0.0	0.0
Erythromycin ^c^	≤0.25–>2			≤0.25	0.5	≥1
*Gemella morbillorum*	≤0.25–>2	≤0.25	>2	72.7	0.0	27.3
GH group	≤0.25–>2	≤0.25	>2	61.1	5.6	33.3
*Gemella taiwanensis*	≤0.25–>2	≤0.25	>2	53.8	0.0	46.2
*Gemella sanguinis*	≤0.25–1	–	–	66.7	0.0	33.3
*Gemella bergeri*	≤0.25	–	–	100.0	0.0	0.0
Total	≤0.25–>2	≤0.25	>2	65.5	1.7	32.8
Clarithromycin	≤0.12–>16					
*Gemella morbillorum*	≤0.12–>16	≤0.12	>16	NA	NA	NA
GH group	≤0.12–16	≤0.12	8	NA	NA	NA
*Gemella taiwanensis*	≤0.12–8	≤0.12	2	NA	NA	NA
*Gemella sanguinis*	≤0.12–0.25	–	–	NA	NA	NA
*Gemella bergeri*	≤0.12	–	–	NA	NA	NA
Total	≤0.12–>16	≤0.12	8	NA	NA	NA
Azithromycin	≤0.12–>4					
*Gemella morbillorum*	≤0.12–>4	≤0.12	>4	NA	NA	NA
GH group	≤0.12–>4	≤0.12	>4	NA	NA	NA
*Gemella taiwanensis*	≤0.12–>4	0.25	>4	NA	NA	NA
*Gemella sanguinis*	0.25–4	–	–	NA	NA	NA
*Gemella bergeri*	0.25	–	–	NA	NA	NA
Total	≤0.12–>4	≤0.12	>4	NA	NA	NA
Clindamycin ^c^	≤0.25–>2			≤0.25	0.5	≥1
*Gemella morbillorum*	≤0.25–>2	≤0.25	>2	72.7	0.0	27.3
GH group	≤0.25–1	≤0.25	0.5	77.8	16.7	5.6
*Gemella taiwanensis*	≤0.25	≤0.25	≤0.25	100.0	0.0	0.0
*Gemella sanguinis*	≤0.25	–	–	100.0	0.0	0.0
*Gemella bergeri*	≤0.25	–	–	100.0	0.0	0.0
Total	≤0.25–>2	≤0.25	>2	82.8	5.2	12.1
Erythromycin/clindamycin	≤1/0.5–>1/0.5					
*G. morbillorum*	≤1/0.5–>1/0.5	≤1/0.5	>1/0.5	NA	NA	NA
GH group	≤1/0.5–>1/0.5	≤1/0.5	≤1/0.5	NA	NA	NA
*G. taiwanensis*	≤1/0.5	≤1/0.5	≤1/0.5	NA	NA	NA
*G. sanguinis*	≤1/0.5	–	–	NA	NA	NA
*Gemella bergeri*	≤1/0.5	–	–	NA	NA	NA
Total	≤1/0.5–>1/0.5	≤1/0.5	>1/0.5	NA	NA	NA
Levofloxacin ^c^	≤0.004–>128			≤2	4	≥8
*Gemella morbillorum*	0.03–>128	0.25	1	90.9	0.0	9.1
GH group	0.125–>128	1	>128	50.0	0.0	50.0
*Gemella taiwanensis*	0.125–>128	>128	>128	38.5	0.0	61.5
*Gemella sanguinis*	0.5–>128	–	–	33.3	0.0	66.7
*Gemella bergeri*	0.5	–	–	100.0	0.0	0.0
Total	0.03–>128	0.5	>128	63.8	0.0	36.2
Moxifloxacin	≤0.5–>2					
*G* *emella morbillorum*	≤0.5–>2	≤0.5	>2	NA	NA	NA
GH group	≤0.5–>2	≤0.5	>2	NA	NA	NA
*Gemella taiwanensis*	≤0.5–>2	>2	>2	NA	NA	NA
*Gemella sanguinis*	≤0.5–>2	–	–	NA	NA	NA
*Gemella bergeri*	≤0.5	–	–	NA	NA	NA
Total	≤0.5–>2	≤0.5	>2	NA	NA	NA
Minocycline	≤1–>8					
*Gemella morbillorum*	≤1–>8	≤1	2	NA	NA	NA
GH group	≤1–8	≤1	8	NA	NA	NA
*Gemella taiwanensis*	≤1–8	≤1	8	NA	NA	NA
*Gemella sanguinis*	≤1	–	–	NA	NA	NA
*Gemella bergeri*	≤1	–	–	NA	NA	NA
Total	≤1	≤1	8	NA	NA	NA
Sulfamethoxazole– trimethoprim	≤9.5/0.5–>38/2					
*Gemella morbillorum*	≤9.5/0.5–>38/2	19/1	>38/2	NA	NA	NA
GH group	≤9.5/0.5–>38/2	38/2	>38/2	NA	NA	NA
*Gemella taiwanensis*	≤9.5/0.5–>38/2	19/1	19/1	NA	NA	NA
*Gemella sanguinis*	19/1–>38/2	–	–	NA	NA	NA
*Gemella bergeri*	≤9.5/0.5	–	–	NA	NA	NA
Total	≤9.5/0.5–>38/2	19/1	>38/2	NA	NA	NA
Gentamicin	≤1–>8					
*Gemella morbillorum*	≤1–8	2	8	NA	NA	NA
GH group	≤1–2	≤1	2	NA	NA	NA
*Gemella taiwanensis*	≤1–4	2	4	NA	NA	NA
*Gemella sanguinis*	≤1–8	–	–	NA	NA	NA
*Gemella bergeri*	2, 4	–	–	NA	NA	NA
Total	≤1–8	2	8	NA	NA	NA
Gentamicin 500	≤500–>500					
*Gemella morbillorum*	≤500	≤500	≤500	NA	NA	NA
GH group	≤500	≤500	≤500	NA	NA	NA
*Gemella taiwanensis*	≤500	≤500	≤500	NA	NA	NA
*Gemella sanguinis*	≤500	–	–	NA	NA	NA
*Gemella bergeri*	≤500	–	–	NA	NA	NA
Total	≤500	≤500	≤500	NA	NA	NA
Arbekacin	≤1–>8					
*Gemella morbillorum*	≤1–8	8	>8	NA	NA	NA
GH group	≤1–8	4	8	NA	NA	NA
*Gemella taiwanensis*	2–>8	4	8	NA	NA	NA
*Gemella sanguinis*	4–>8	–	–	NA	NA	NA
*Gemella bergeri*	4, >8	–	–	NA	NA	NA
Total	≤1–8	4	>8	NA	NA	NA
Fosfomycin	≤16–>128					
*G* *emella morbillorum*	≤16–32	≤16	≤16	NA	NA	NA
GH group	≤16	≤16	≤16	NA	NA	NA
*Gemella taiwanensis*	≤16	≤16	≤16	NA	NA	NA
*Gemalla sanguinis*	≤16	–	–	NA	NA	NA
*Gemella bergeri*	≤16	–	–	NA	NA	NA
Total	≤16–32	≤16	≤16	NA	NA	NA
Rifampicin	≤0.5–>2					
*Gemella morbillorum*	≤0.5	≤0.5	≤0.5	NA	NA	NA
GH group	≤0.5	≤0.5	≤0.5	NA	NA	NA
*Gemella taiwanensis*	≤0.5	≤0.5	≤0.5	NA	NA	NA
*Gemella sanguinis*	≤0.5	–	–	NA	NA	NA
*Gemella bergeri*	≤0.5			NA	NA	NA
Total	≤0.5	≤0.5	≤0.5	NA	NA	NA
Vancomycin ^c^	≤0.25–>2			≤1		
*Gemella morbillorum*	≤0.25–0.5	0.5	0.5	100.0	0.0	0.0
GH group	≤0.25–0.5	0.5	0.5	100.0	0.0	0.0
*Gemella taiwanensis*	≤0.25–0.5	0.5	0.5	100.0	0.0	0.0
*Gemella sanguinis*	≤0.25–0.5	–	–	100.0	0.0	0.0
*Gemella bergeri*	0.5	–	–	100.0	0.0	0.0
Total	≤0.25–0.5	0.5	0.5	100.0	0.0	0.0
Teicoplanin	≤0.5–>16					
*Gemella morbillorum*	≤0.5	≤0.5	≤0.5	NA	NA	NA
GH group	≤0.5	≤0.5	≤0.5	NA	NA	NA
*Gemella taiwanensis*	≤0.5	≤0.5	≤0.5	NA	NA	NA
*Gemella sanguinis*	≤0.5	–	–	NA	NA	NA
*Gemella bergeri*	≤0.5	–	–	NA	NA	NA
Total	≤0.5	≤0.5	≤0.5	NA	NA	NA
Linezolid	≤0.5–>4					
*Gemella morbillorum*	≤0.5–1	≤0.5	1	NA	NA	NA
GH group	≤0.5–1	≤0.5	1	NA	NA	NA
*Gemella taiwanensis*	≤0.5	≤0.5	≤0.5	NA	NA	NA
*Gemella sanguinis*	≤0.5–1	–	–	NA	NA	NA
*Gemella bergeri*	≤0.5, 2	–	–	NA	NA	NA
Total	≤0.5	≤0.5	1	NA	NA	NA
Daptomycin	≤0.25–>4					
*Gemella morbillorum*	≤0.25–4	2	2	NA	NA	NA
GH group	0.5–2	1	2	NA	NA	NA
*Gemella taiwanensis*	≤0.25–2	1	2	NA	NA	NA
*Gemella sanguinis*	1–4	–	–	NA	NA	NA
*Gemella bergeri*	2, 4	–	–	NA	NA	NA
Total	≤0.25–4	1	2	NA	NA	NA

^a^ Interpretive breakpoints are shown in bold for each antibiotic. ^b^ NA, not applicable (breakpoints not established). ^c^ Antimicrobial agents with breakpoints listed in CLSI M45-third edition.

**Table 3 antibiotics-12-01538-t003:** Distribution of macrolides and clindamycin MICs and possession of the *mef*, *erm,* and *msrA* genes in erythromycin-non-susceptible *Gemella* isolates.

Strain No.	Identification	MIC (μg/mL)					Macrolide Phenotype ^a,b^	*mefA/E*	*erm*	*msrA*
		Erythromycin	Clindamycin	Erythromycin/Clindamycin	Clarithromycin	Azithromycin				
TWCC 57201	*Gemella morbillorum*	>2	>2	>1/0.5	8	>4	cMLS_B_	-	*ermB*	*-*
TWCC 57818	*Gemella morbillorum*	>2	>2	>1/0.5	>16	>4	cMLS_B_	-	*ermB*	*-*
TWCC 57944	*Gemella morbillorum*	>2	>2	>1/0.5	>16	>4	cMLS_B_	-	*ermB*	*-*
TWCC 59111	*Gemella morbillorum*	>2	>2	>1/0.5	8	>4	cMLS_B_	-	*ermB*	*-*
TWCC 71703	*Gemella morbillorum*	>2	>2	>1/0.5	>16	>4	cMLS_B_	-	*ermB*	*-*
TWCC 72266	*Gemella morbillorum*	>2	>2	>1/0.5	>16	>4	cMLS_B_	-	*ermB*	*-*
TWCC 52027	GH group	0.5	≤0.25	≤1/0.5	8	2	M	*mefE*	-	-
TWCC 59566	GH group	2	≤0.25	≤1/0.5	2	>4	M	*mefE*	-	-
TWCC 59567	GH group	>2	1	>1/0.5	16	>4	M	*mefE*	-	-
TWCC 59795	GH group	1	0.5	≤1/0.5	0.5	2	M	*mefE*	-	-
TWCC 70939	GH group	>2	≤0.25	≤1/0.5	2	>4	M	*mefE*	-	-
TWCC 71200	GH group	1	≤0.25	≤1/0.5	2	2	M	*mefA*	-	-
TWCC 71814	GH group	1	≤0.25	≤1/0.5	0.5	1	M	*mefE*	-	-
TWCC 55344	*Gemella taiwanensis*	>2	≤0.25	≤1/0.5	8	>4	M	*mefE*	-	-
TWCC 58522	*Gemella taiwanensis*	>2	≤0.25	≤1/0.5	2	4	M	*mefE*	-	-
TWCC 70386	*Gemella taiwanensis*	>2	≤0.25	≤1/0.5	2	4	M	*mefE*	-	-
TWCC 72085	*Gemella taiwanensis*	>2	≤0.25	≤1/0.5	2	>4	M	*mefE*	-	-
TWCC 70387L	*Gemella taiwanensis*	2	≤0.25	≤1/0.5	0.5	>4	M	*mefE*	-	-
TWCC 70387S	*Gemella taiwanensis*	>2	≤0.25	≤1/0.5	2	>4	M	*mefE*	-	-
TWCC 54965	*Gemella sanguinis*	1	≤0.25	≤1/0.5	0.25	4	M	*mefE*	-	-
TWCC 70419	*Gemella sanguinis*	≤0.25 ^c^	≤0.25	≤1/0.5	≤0.12	0.25	not M	*mefE*	-	-

^a^ cMLSB: macrolide–lincosamide–streptogramin B-resistant phenotype. ^b^ M: macrolide-resistant phenotype. ^c^ Erythromycin-susceptible.

**Table 4 antibiotics-12-01538-t004:** Distribution of minocycline MIC and *ermB* in *Gemella* isolates harboring the *tetM* gene.

Strain No.	Identification	*tetM*	Minocycline MIC (μg/mL)	*ermB*
TWCC 57944	*Gemella morbillorum*	+	2	+
TWCC 57987	*Gemella morbillorum*	+	≤1	−
TWCC 59111	*Gemella morbillorum*	+	2	+
TWCC 70937	*Gemella morbillorum*	+	>8	−
TWCC 71703	*Gemella morbillorum*	+	2	+
TWCC 72266	*Gemella morbillorum*	+	≤1	+
TWCC 51800	GH group	+	8	−
TWCC 59795	GH group	+	≤1	−
TWCC 70939	GH group	+	8	−
TWCC 71814	GH group	+	2	−
TWCC 53044	*Gemella taiwanensis*	+	8	−
TWCC 56546	*Gemella taiwanensis*	+	2	−
TWCC 58522	*Gemella taiwanensis*	+	8	−
TWCC 70386	*Gemella taiwanensis*	+	8	−
TWCC 72085	*Gemella taiwanensis*	+	8	−
TWCC 70387L	*Gemella taiwanensis*	+	≤1	−
TWCC 70387S	*Gemella taiwanensis*	+	≤1	−

**Table 5 antibiotics-12-01538-t005:** Distribution of MIC of tested quinolones and amino acid substitutions in *gyrA* gene in quinolone-resistant *Gemella* isolates.

Strain	*n*	MIC (μg/mL)		GyrA Amino Acid Substitutions ^a^
		Levofloxacin	Moxifloxacin	
*Gemella morbillorum*	2	>128	>2	Ser83 > Leu83 (*n* = 2)
GH group	9	128–>128	>2	Ser83 > Phe83 (*n* = 7), Ser83 > Tyr83 (*n* =2)
*Gemella taiwanensis*	8	128–>128	>2	Ser83 > Phe83 (*n* = 7), Ser83 > Tyr83 (*n* = 1)
*Gemella sanguinis*	2	128–>128	>2	Ser83 > Phe83 (*n* = 2)

^a^ *gyrA*-Ser83 Leu: serine to leucine at codon 83; Ser83 Phe: serine to phenylalanine at codon 83; Ser83 Tyr; serine to tyrosine at codon 83.

## Data Availability

The data that support the findings of this study are available from the corresponding author KK upon reasonable request.

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
