# Peer review of "Antimicrobial Susceptibility to 27 Drugs and the Molecular Mechanisms of Macrolide, Tetracycline, and Quinolone Resistance in Gemella sp."

_antibiotics, 2023, doi:10.3390/antibiotics12101538_

Round 1
Reviewer 1 Report
The authors have conducted extensive work on antimicrobial susceptibility to 27 drugs and molecular mechanisms of macrolide, tetracycline, and quinolone resistance in Gemella sp.
The text is well written, the results are presented clearly and the conclusions are in accordance with the objective of the work. The bibliography is current and complete. References from the last 10 years reach 45%.
Some minor observations are suggested:
1-Page 3, line 129: MIC90 must have the number 90 in subscript
2-Limitations of the study are missing
3-Quote 46 is incomplete
1-Page 5, line 201: it says “endemic sites” and perhaps, it could be changed to body sites, usual colonization sites, body areas, or similar
Author Response
Point-by-Point Response to Reviewer 1
The authors have conducted extensive work on antimicrobial susceptibility to 27 drugs and molecular mechanisms of macrolide, tetracycline, and quinolone resistance in Gemella sp.
The text is well written, the results are presented clearly and the conclusions are in accordance with the objective of the work. The bibliography is current and complete. References from the last 10 years reach 45%.
Some minor observations are suggested:
Thank you for your understanding on our study.
According to your comments, we revised our manuscript.
1-Page 3, line 129: MIC90 must have the number 90 in subscript
The word was corrected.
2-Limitations of the study are missing
According to your suggestion, we described Limitation at the bottom of Discussion as below.
“This study has some limitations. We did not collect patient information, such as clinical history, antibiotics used for treatment, prognosis, isolated hospitals, and time period of collection. To further analyze the characteristics of Gemella species, patient in-formation might be helpful.”
(Lines 292-295)
3-Quote 46 is incomplete
The quote was corrected.
Comments on the Quality of English Language
1-Page 5, line 201: it says “endemic sites” and perhaps, it could be changed to body sites, usual colonization sites, body areas, or similar
According to your suggestion, we replaced “endemic sites” with “usual colonization sites”. (Line 209)
Reviewer 2 Report
The manuscript surveys the susceptibility of 58 Gemella strains to 27 antimicrobial drugs according to protocol CLSI M45-A3. Screening of genes known to be responsible for AMR were undertaken so that mechanisms for susceptibility to drug classes for each strain could be suggested. Overall, this manuscript could be helpful to those who study Gemella and wish to have a thorough overview of its susceptibility to various drugs. Provided are my suggestions for improvement.
There are some issues with formatting and consistency that can be easily fixed. For example, the abstract doesn't need to be broken up into sections like Background, Methods, Results. There also needs to be consistency with using or not using abbreviations; for example, you use MLSA without defining it but a few sentences beforehand that spell out the full method name for MALDI and don't provide the abbreviation. Consistency with use of abbreviations would be neater.
Line 77, Do these numbers correspond respectively to these bacteria?
Line 101 how are you defining "no significant difference"? The ratios reported vary from 0-53%, which seems significant. Can you report a P value for this set (like you did for the other data sets)?
Given the nature of the manuscript, it would be preferred that the methodology cited for [55] is explicitly described. Researchers who would be interested in this paper may want to use your use your methods directly and it would be better for them to have the protocol you used directly accessible instead of needing to dig for the protocol (granted, the citation is from this same journal).
Author Response
Point-by-Point Response to Reviewer 2
Comments and Suggestions for Authors
The manuscript surveys the susceptibility of 58 Gemella strains to 27 antimicrobial drugs according to protocol CLSI M45-A3. Screening of genes known to be responsible for AMR were undertaken so that mechanisms for susceptibility to drug classes for each strain could be suggested. Overall, this manuscript could be helpful to those who study Gemella and wish to have a thorough overview of its susceptibility to various drugs. Provided are my suggestions for improvement.
Thank you for your understanding on our study.
There are some issues with formatting and consistency that can be easily fixed. For example, the abstract doesn't need to be broken up into sections like Background, Methods, Results. There also needs to be consistency with using or not using abbreviations; for example, you use MLSA without defining it but a few sentences beforehand that spell out the full method name for MALDI and don't provide the abbreviation. Consistency with use of abbreviations would be neater.
According to your comments, we revised our manuscript. We removed “(1) Background, (2) Methods, (3) Results, (4) Conclusion. Definition of the word MLSA was added. Thank you for your kind teaching.
Line 77, Do these numbers correspond respectively to these bacteria?
Yes. To show clearly that these numbers correspond respectively to these bacteria, we inserted “respectively” as follows; “In total, 4, 3, and 12 isolates were assigned to G. haemolysans, G. parahaemolysans, and G. taiwanensis, respectively, both by 16S rDNA sequencing and MLSA”
(Lines 76-77)
Line 101 how are you defining "no significant difference"? The ratios reported vary from 0-53%, which seems significant. Can you report a P value for this set (like you did for the other data sets)?
As you pointed out, the ratios varied from 0 to 53%. However, number of G. sanguinis (3 strains) and G. bergeri (2 strains) was too small to compare. There was no significant difference among G. morbillorum, GH group, and G. taiwanensis.
Given the nature of the manuscript, it would be preferred that the methodology cited for [55] is explicitly described. Researchers who would be interested in this paper may want to use your use your methods directly and it would be better for them to have the protocol you used directly accessible instead of needing to dig for the protocol (granted, the citation is from this same journal).
According to the reviewer’s suggestion, we described the methods as follows; “Briefly, the genes were amplified using the TaKaRa Ex Premier DNA Polymerase (TaKaRa Bio) and pairs of primers (Table S2) with the following thermal cycle: initial denaturation at 94°C for 5 min; 30 cycles of denaturation at 98°C for 10 s, primer annealing at the indicated temperatures (Table S2) for 15 s, and extension at 68°C for 70 s; and final extension at 72°C for 5 min.”
(Lines 344-348).
Reviewer 3 Report
This manuscript discusses the antimicrobial susceptibility of 52 clinical isolates of Gemella sp. to 27 antimicrobial drugs and molecular mechanisms of drug resistance specifically against three class of antibiotics. The research reported on this pathogen is an important contribution in clinical research. However, the manuscript have shortcomings which are mentioned as below:
Major comments:
1) 4.1. Bacterial strains: There is no mention of the hospital or healthcare facility from where the isolates were isolated? What was the time period during which isolation was carried out? What about the clinical history (previous or ongoing antibiotic therapy) of patients/subjects?
2) Line 32: Rephrase Conclusion part in abstract…Several antimicrobial resistances??
3) How antimicrobial susceptibility of bacterial isolates was evaluated. Authors have just referred to CLSI M45-A3 but the brief procedure should be included in relevant sub-section of materials and methods.
4) Line 290-293: Authors should specify that a particular type strain was obtained from that specific culture collection/provider. These details are missing.
5) Line 334; section 4.7: Authors should first mention the name of antibiotic and then write its short code in brackets. For example: Cefotaxime (CTX),….so on……………………
6) Line 365: Rephrase the Table 1 caption ‘List of origins of Gemella isolated from clinical specimens’
7) Line 287; 4.1: Authors have mentioned “We collected 51 Gemella strains isolated from clinical sites” in section 4.1 whereas in Table 1 and other instances in text, the number of isolates are 52. Is it just a typo error or something is miscalculated?
8) Why authors are giving two data sets, one for Gentamicin and other for Gentamicin500? What is the rationale?
9) MIC50 and MIC90 are almost same in all isolates for nearly all the tested antimicrobials. Any explanation?
10) Discussion, line 188 onwards: For the sake of ease of understanding and interpretation of readers, it is suggested that antibiotic names should be written rather than their codes.
11) Authors should check the expression “CLSI M45-A3”. For what A3 stands? Does it represent 3rd edition?
12) Section 4.7: There is no mention of standard control strains used in AST experiments. Have you used S. pneumoniae ATCC® 49619 and other CLSI mandated strains as standard? Authors should include their AST data also in tables/text.
13) Table 2: It is not clear why authors have tested those antimicrobials where CLSI interpretation criteria were not available? In this case, it is uncertain how to classify the isolates as S, I or R. What other most approximate and relevant interpretation criteria can be relied upon?
14) Authors should include a figure/bar diagram indicating percentage of isolates which are resistant or sensitive to each antibiotics/combination. Similarly, an easy-to-interpret table/figure/heatmap may be included which depict the least resistant and most resistant isolates out of n=52 isolates.
15) There are multiple English grammar issues with the manuscript. It is suggested to get it checked by an expert.
16) Refine & rephrase figure & table captions/title/legends
Minor comments:
(a) Line 423: Reference 1 title is written in CAPITAL letters. Correct it
(b) Line 533: Reference 46 lack the name of the journal.
(c) Many references are incomplete as volume, page no., journal names are missing.
(d) Please ensure that the scientific names should be italicized within reference list/bibliography also.
Considering the merit and strengths of the research work embodied in the manuscript, I am of opinion that the manuscript needs considerable revision before it can be considered for publication.
There are multiple English grammar issues with the manuscript. It is suggested to get it checked by an expert.
Author Response
Point-by-Point Response to Reviewer 3
Comments and Suggestions for Authors
This manuscript discusses the antimicrobial susceptibility of 52 clinical isolates of Gemella sp. to 27 antimicrobial drugs and molecular mechanisms of drug resistance specifically against three class of antibiotics. The research reported on this pathogen is an important contribution in clinical research. However, the manuscript have shortcomings which are mentioned as below:
Thank you very much for your thoughtful comments. According to your comments and suggestions, we revised our manuscript.
Major comments:
1) 4.1. Bacterial strains: There is no mention of the hospital or healthcare facility from where the isolates were isolated? What was the time period during which isolation was carried out? What about the clinical history (previous or ongoing antibiotic therapy) of patients/subjects?
As you pointed out, we did not mention of information on patients/subjects. We mentioned the absence in the Discussion as follows; “This study has some limitations. We did not collect patient information, such as clinical history, antibiotics used for treatment, prognosis, isolated hospitals, and time period of collection. To further analyze the characteristics of Gemella species, patient information might be helpful.”
(Lines 292-295)
2) Line 32: Rephrase Conclusion part in abstract…Several antimicrobial resistances
We rephrased the Conclusion part in abstract as follows; “The mechanisms of resistance to erythromycin and clindamycin differed across Gemella species. In addition, the rate of susceptibility to levofloxacin differed across Gemella sp., and the quinolone resistance mechanism was caused by mutations in GyrA alone.”
(Lines 31-34)
3) How antimicrobial susceptibility of bacterial isolates was evaluated. Authors have just referred to CLSI M45-A3 but the brief procedure should be included in relevant sub-section of materials and methods.
According to the reviewer’s suggestion, we described the method for antimicrobial susceptibility test as follows: “Briefly, Gemella strains grown on 5% sheep blood agar EX plates (Shimadzu Diagnostics Corporation, Tokyo, Japan) were suspended in saline and inoculated in Difco cation-adjusted Mueller Hinton Broth (Becton, Dickinson, Sparks, MD, USA) supplemented with lysed 5% horse blood (KOHJIN BIO, Saitama, Japan) at 5 × 105 CFU/mL. Dry plates 34, 42, and 44 were purchased from Eiken Chemical Co., Ltd. (Tokyo, Japan). Dry plates were incubated at 35°C and 5% CO2 for 48 h. In cases of poor growth until 48 h, the strains were cultured for 72–96 h. MICs were interpreted using the CLSI M45 breakpoints for Gemella spp. when available [32]. For quality control, CLSI-3rd Edition-recommended Streptococcus pneumoniae ATCC 49619 was used [32]. Quality control was performed for each change of plates in every lot.”
(Lines 362-371)
4) Line 290-293: Authors should specify that a particular type strain was obtained from that specific culture collection/provider. These details are missing.
According to your suggestion, we revised explanation regarding the type strains as follows: “G. morbillorum ATCC 27824T, and G. haemolysans ATCC 10379T were obtained from American Type Culture Collection Manassas, VA, USA, G. parahaemolysans JCM 18067T, and G. taiwanensis JCM 18066T, were collected from the Japan Collection of Microorganisms, RIKEN BRC, Ibaraki, Japan. G. sanguinis CCUG 37820T and G. bergeri CCUG 37817T were obtained from the Culture Collection University of Götheborg, Götheborg, Sweden.”
(Lines 304-309)
5) Line 334; section 4.7: Authors should first mention the name of antibiotic and then write its short code in brackets. For example: Cefotaxime (CTX),….so on……………………
According to another reviewer’s suggestion below (10), we replaced all the short codes with the names such as erythromycin and cefotaxime.
6) Line 365: Rephrase the Table 1 caption ‘List of origins of Gemella isolated from clinical specimens’
We rephrased the caption with “ Isolated sites of Gemella species used in this study.”
7) Line 287; 4.1: Authors have mentioned “We collected 51 Gemella strains isolated from clinical sites” in section 4.1 whereas in Table 1 and other instances in text, the number of isolates are 52. Is it just a typo error or something is miscalculated?
The number of strains from clinical specimens is 52. We have corrected the number. We sincerely appreciate your perceptive remarks.
8) Why authors are giving two data sets, one for Gentamicin and other for Gentamicin500? What is the rationale?
Streptococci are usually aminoglycoside resistant. Thus, we also tested GM500 examined to detect Gemella strains which are highly resistant to aminoglycoside.
9) MIC50 and MIC90 are almost same in all isolates for nearly all the tested antimicrobials. Any explanation?
Because the most Gemella strains used in our study were highly sensitive to the β-lactams, MIC50 and MIC90 of the drugs became close or same.
(Lines 90-93)
10) Discussion, line 188 onwards: For the sake of ease of understanding and interpretation of readers, it is suggested that antibiotic names should be written rather than their codes.
According to another reviewer’s suggestion below (10), we replaced all the short codes with the names such as erythromycin and cefotaxime.
11) Authors should check the expression “CLSI M45-A3”. For what A3 stands? Does it represent 3rd edition?
As the reviewer pointed out, CLSI M45-3rd Edition is correct.
12) Section 4.7: There is no mention of standard control strains used in AST experiments. Have you used S. pneumoniae ATCC® 49619 and other CLSI mandated strains as standard? Authors should include their AST data also in tables/text.
According to the reviewer’s suggestion, we added a sentence as follows :“For quality control, CLSI-3rd Edition-recommended Streptococcus pneumoniae ATCC 49619 was used [32]. Quality control was performed for each change of plates in every lot.”
(Lines 370-371)
The AST values were slightly varied across the lots, we did not include the data.
13) Table 2: It is not clear why authors have tested those antimicrobials where CLSI interpretation criteria were not available? In this case, it is uncertain how to classify the isolates as S, I or R. What other most approximate and relevant interpretation criteria can be relied upon?
As you pointed out, we cannot evaluate sensitivity (S, I, or R) for drugs whose breakpoints are not described in CLSI. Nonetheless, we believe that the MIC50 and MIC90 give some knowledge to understand the characteristics of Gemella isolates.
14) Authors should include a figure/bar diagram indicating percentage of isolates which are resistant or sensitive to each antibiotics/combination. Similarly, an easy-to-interpret table/figure/heatmap may be included which depict the least resistant and most resistant isolates out of n=52 isolates.
According to the reviewer’s suggestion, we prepared two figures. Figures 1 and 2 show ratios of the resistant strains and distribution of MIC, respectively. We believe these figures help the readers to understand our data.
15) There are multiple English grammar issues with the manuscript. It is suggested to get it checked by an expert.
According to the reviewer’s suggestion, the revised manuscript was proofed again by English experts.
16) Refine & rephrase figure & table captions/title/legends
The titles and captions were revised.
Minor comments:
(a) Line 423: Reference 1 title is written in CAPITAL letters. Correct it
The quote was corrected.
(b) Line 533: Reference 46 lack the name of the journal.
(c) Many references are incomplete as volume, page no., journal names are missing.
(d) Please ensure that the scientific names should be italicized within reference list/bibliography also.
We thoroughly checked the references.
Considering the merit and strengths of the research work embodied in the manuscript, I am of opinion that the manuscript needs considerable revision before it can be considered for publication.
We would like to express our sincere appreciation for the reviewer’s peer review.
Comments on the Quality of English Language
There are multiple English grammar issues with the manuscript. It is suggested to get it checked by an expert.
As described above, the revised manuscript was proofed again by English experts.
Reviewer 4 Report
The work presented in the manuscript entitled "Antimicrobial Susceptibility to 27 Drugs and Molecular Mechanisms of Macrolide, Tetracycline, and Quinolone Resistance in Gemella sp." is an interesting addition in clinical sciences regarding the antimicrobial profiles of generally less known bacterium. Particularly the 16S rDNA and MLST analysis of the isolates are standard tools described in the current investigation. Please see below one query:
1. In the methods section: Line 287-290, the ambiguity in total number of isolates/ strains. Is there 51 or 52 isolated strains along with 5 or 6 type strains.
Thanks and Regards
Author Response
Point-by-Point Response to Reviewer 4
Comments and Suggestions for Authors
The work presented in the manuscript entitled "Antimicrobial Susceptibility to 27 Drugs and Molecular Mechanisms of Macrolide, Tetracycline, and Quinolone Resistance in Gemella sp." is an interesting addition in clinical sciences regarding the antimicrobial profiles of generally less known bacterium. Particularly the 16S rDNA and MLST analysis of the isolates are standard tools described in the current investigation. Please see below one query:
Thank you for your understanding our study.
- In the methods section: Line 287-290, the ambiguity in total number of isolates/ strains. Is there 51 or 52 isolated strains along with 5 or 6 type strains.
The number of strains from clinical specimens is 52. We have corrected the number.
Round 2
Reviewer 3 Report
The revised manuscript is considerably improved and all the queries have been addressed by the authors to a satisfactory extent.
I am pleased to recommend the manuscript for the publication.